# Antidepressive Effect of Natural Products and Their Derivatives Targeting BDNF-TrkB in Gut–Brain Axis

**DOI:** 10.3390/ijms232314968

**Published:** 2022-11-29

**Authors:** Humna Liaqat, Amna Parveen, Sun-Yeou Kim

**Affiliations:** 1Department of Animal Science, Biotechnical Faculty, University of Ljubljana, Groblje 3, 1230 Domzale, Slovenia; 2College of Pharmacy, Gachon University, No. 191, Hambakmoero, Yeonsu-gu, Incheon 21936, Republic of Korea

**Keywords:** BDNF, depression, TrkB, gut–brain axis, signaling pathways, natural products, microbiome

## Abstract

Modern neurological approaches enable detailed studies on the pathophysiology and treatment of depression. An imbalance in the microbiota–gut–brain axis contributes to the pathogenesis of depression. This extensive review aimed to elucidate the antidepressive effects of brain-derived neurotrophic factor (BDNF)-targeting therapeutic natural products and their derivatives on the gut–brain axis. This information could facilitate the development of novel antidepressant drugs. BDNF is crucial for neuronal genesis, growth, differentiation, survival, plasticity, and synaptic transmission. Signaling via BDNF and its receptor tropomyosin receptor kinase B (TrkB) plays a vital role in the etiopathogenesis of depression and the therapeutic mechanism of antidepressants. This comprehensive review provides information to researchers and scientists for the identification of novel therapeutic approaches for neuropsychiatric disorders, especially depression and stress. Future research should aim to determine the possible causative role of BDNF-TrkB in the gut–brain axis in depression, which will require further animal and clinical research as well as the development of analytical approaches.

## 1. Introduction

The “gut–brain axis” comprises connections including multiple biological systems that allow bidirectional communication between the gut and brain, and it is vital for maintaining homeostasis of the gastrointestinal, microbial, and central nervous systems (CNS). Cross-communication in these biological networks involves direct and indirect signaling via chemical transmitters, neuronal pathways, and the immune system. Considering that multiple biological systems are involved, it is probable that manifold mechanisms and pathways act together to mediate various aspects of disease pathology; further research is needed to understand the underlying mechanisms involved [1]. The gut microbiota acts as a biological rheostat for sensing, modifying, and tuning large amounts of chemical signals from the environment, which then circulate throughout the body. Additionally, gut microbiota communities depict the intersection of the host and the environment, which may directly impact human health, as certain animal behaviors appear to correlate with gut bacterial composition and disturbances in microbial communities have been implicated in several neurological disorders [2].

The World Health Organization (WHO) suggests that depression will be the primary contributor to the global disease burden by 2030 [3,4]. Depression is a blight on the human condition and the root cause of several metabolic disorders. It accounts for more ‘years lost’ to disability than any other condition worldwide owing to the large number of people suffering from it (approximately 350 million) [3]. However, the pathophysiology of depression is poorly understood and is generally undiagnosed owing to stigma, the lack of effective therapies, and inadequate mental health resources. Almost half of the world’s population lives in a country with only two psychiatrists per 100,000 people [5]. In past decades, unraveling the pathophysiology of depression was a unique challenge with respect not only to depressive syndromes but also to their heterogeneous etiologies. Nevertheless, in recent years, a plethora of scientific studies have produced important evidence regarding signaling pathways, novel synaptic repair therapeutics for neurodegeneration and behavioral disorders [6], the molecular mechanisms of depression [7], and its emerging role in the gut–brain axis [8].

In the brain and peripheral tissues, neurotrophins are growth factors; they include brain-derived neurotrophic factor (BDNF), nerve growth factor (NGF), neurotrophin-3, and neurotrophin-4, which regulate neuronal function and help in the development and maintenance of brain function. Alterations in the production of these neurotrophins may lead to pathological conditions such as depression. Therefore, boosting the production of these neurotrophins could potentially serve as an alternative therapeutic strategy for depression. Recent studies have indicated that BDNF is the most important representative and an attractive factor associated with depression.

BDNF, or abrineurin, is a pleiotropic protein encoded by the BDNF gene, which is a member of the neurotrophic factor family that affects the central nervous system as well as the endocrine and immune systems. BDNF is active at the synapses and plays an important role in supporting neuronal survival and in promoting the growth and maturation of new cells [9]. The binding of BDNF to its receptor stimulates neuronal survival in the adult brain and regulates inhibitory and excitatory synaptic transmission and activity-dependent plasticity [10]. The ‘neurotrophin hypothesis of depression’ is mainly based on evidence that antidepressant treatment improves BDNF expression, while a reduction in hippocampal BDNF levels is correlated with stress-induced depressive behaviors [11,12,13,14].

The underlying pathophysiological mechanism of depression remains unknown, which is the main challenge. A new perception targeting the gut microbiota–brain axis might be closely related to the truth in depression and could contribute to the development of novel antidepressant drugs. The fight against depression can be undertaken by either exploring the mechanism for boosting BDNF production or by identifying new potential antidepressant agents that can either substitute for BDNF or increase BDNF levels in the brain.

Although we have accumulated considerable knowledge regarding depression, we need to address the gaps. Through this comprehensive review, we aim to provide a better understanding of the mechanisms of action of neuroprotective natural products and their derivatives targeting BDNF in the gut–brain axis. This information can be used to develop various natural therapeutic agents from medicinal plants, prebiotics, probiotics, dietary supplements, herbal formulations, bodily fluids, and other agents that can potentially interact with the causative factors of abnormalities related to the gut–brain axis and confer neuroprotection. In addition, we highlight the necessity for future studies on the clinical application of these natural products and their metabolites against depression via modulation of the gut–brain axis. 

## 2. Results

### 2.1. Brain-Derived Neurotrophic Factor

BDNF, one of the major neurotrophic factors and a widely studied neurotrophin, is positioned on chromosome 11p14.1. It is initially synthesized in the endoplasmic reticulum as a 32–35 kDa precursor protein (pro-BDNF).

BDNF plays diverse roles in the pathogenesis of depression, depending on the brain region and the individual circuits. It exerts an antidepressant effect in the prefrontal cortex and hippocampus and a prodepressant effect in the mesolimbic dopamine circuit [15]. It is crucial for neuronal genesis, survival, growth, maturation, plasticity, synaptic viability, and synaptic transmission. BDNF can serve as a synaptic repair therapy, as it controls all three aspects of synaptic physiology. It secures and maintains existing synapses and regulates new synapse formation, even in the presence of various toxins [16]. BDNF plays a considerable role in cognitive functioning and mood phases [17].

Prominent levels of BDNF mRNA are detectable in the hippocampus and in specific regions of the cerebellum and cortex of the CNS. Low levels are found in the spinal cord, heart, lung, skeletal muscle, testis, prostate, and placenta [18]. A smaller amount exists in the eye, skin, ovaries, and kidney [19]. Physical exercise, dietary restrictions, and antidepressants are known strategies for increasing BDNF levels in the brain [10,20]. Insufficient BDNF is a leading risk factor for impaired neuroplasticity and depressive symptoms [21]. Several studies have demonstrated that patients with depression have lower BDNF levels than healthy individuals [22].

### 2.2. Signaling Pathway of BDNF

BDNF secretion is activity-dependent and is secreted at different stimulation intensities by presynaptic and postsynaptic terminals. BDNF can bind to p75 neurotrophin receptors with a lower affinity and to TrkB receptors with a higher affinity [23]. BDNF transcripts may produce the specific regional and temporal effects of BDNF; they are a focus of active investigation. Activity-dependent BDNF transcription can be modulated by epigenetic alterations in the chromatin structure [24]. The transcription of BDNF mRNA can be controlled by Ca^2+^ influx via Ca^2+^-permeable NMDA receptors and voltage-gated Ca^2+^ channels. Previous research has demonstrated that Ca^2+^ commences the binding of transcription factors such as Ca^2+^ response factor (CaRF) and cyclic AMP response element binding protein (CREB) to BDNF promoters [25].

TrkB receptors exist at the pre- and postsynapses. TrkB phosphorylation induced by BDNF binding to the TrkB receptor can regulate a minimum of three intracellular cascades: phospholipase C-γ (PLC-γ), MAPK/extracellular signal-regulated kinase (ERK), and PI3K/AKT signaling [26]. BDNF binds to TrkB, thus initiating receptor dimerization, the stimulation of tyrosine kinase activity, and the phosphorylation of specific tyrosine residues, which creates docking sites for adaptors that couple these receptors to intracellular signaling cascades [27]. A study revealed that the BDNF–TrkB pathway is mediated through *N*-methyl-aspartate receptor (NMDAR)–calcium/calmodulin-dependent protein kinase II (CaMKII) signaling and BDNF release in the dendritic spine, manifesting the vital role of BDNF and TrkB in structural and functional plasticity [28]. NMDA glutamate receptor dependent long-term potentiation (LTP) is associated with synaptic plasticity, which is the ability of neurons to change connections between neuronal networks in response to use or disuse. BDNF increases LTP and synaptic plasticity by increasing NMDA receptor phosphorylation [29,30]. In contrast, NMDA induces BDNF synthesis, which contributes to synaptic remodeling and prevents glutamate toxicity mediated through TrkB receptor activation [31]. In addition, the reduction in synaptic density and proteins (in response to depression) in the prefrontal cortexes of patients with depression is associated with the upregulation of a transcriptional repressor of synaptic proteins [32]. Insights into the signaling pathway of BDNF are presented in Figure 1.

Hence, stress and depression disturb BDNF–TrkB receptor signaling, along with the depletion of the downstream ERK and Akt pathways in the hippocampus and prefrontal cortex [33]. These pathways are positively involved in synaptic maturation and stability via the stimulation of synaptic protein synthesis and glutamate receptor cycling [34]. Decreased Akt activity in dopamine neurons is linked to increased susceptibility to stress and is reversed by antidepressant treatment in rodents and human postmortem tissue [35,36]. Furthermore, depression and stress enhance the levels of a negative regulator of ERK signaling, MAPK phosphatase-1, which causes depressive behaviors and reduces the levels of the downstream target of BDNF, which is necessary for normal antidepressant activity [37].

### 2.3. Role of Gut Microbiota in Depression

The correlative impact of the gastrointestinal tract (GIT) on brain function has been acknowledged for half of the nineteenth century through pioneering studies. A dynamic community of complex microbiota in the human intestine has 10 times more microorganisms than it has human cells [38]. Studies have claimed that the gut microbiota interacts with the CNS through several routes, further deepening our understanding of its impact on mental health, including depression [2,39].

The bidimensional relationship of the gut–brain axis is regulated through different neuroactive modulators such as BDNF and serotonin (5-HT) and signaling molecules including lipopolysaccharides, tryptophan metabolites, and trimethylamine-*N*-oxide (TMAO), which are involved in health and disease [40]. These chemical signals are produced by specific gut microbiota that have precise roles in many biological processes related to the gut–brain axis in health and disease.

Gut microbes are vital to the human body, and the microbiota composition is highly individual and regulated by growth, genes, dietary habits, medication exposure, development, and location. The brain and gut are operational in a two-directional manner; they can influence each other’s roles and remarkably affect anxiety, stress, depression, and cognition [41]. Several pathways have been implicated in inter-related communication. Research has shown that a healthy gut microbiota transports brain signals through various pathways, including microglial activation, neurogenesis, neural transmission, and behavioral control under stable or stressful events. This process indicates the significant role of microbiomes in managing mental health issues. Depression additionally involves the dysregulation of the neuroendocrine and neuroimmune pathways. The intestinal microbiota plays a vital role in structuring, establishing, and sustaining mental health during the intrauterine period. Microbiota strategies are important for understanding and treating the pathogenesis of neuropsychiatric disorders, especially depression. Antidepressant drugs, which are the most obvious option for the treatment of depression, also produce antimicrobial and immunomodulatory effects [8].

#### 2.3.1. Gut Microbiota–Brain Axis in Depression

Stress, anxiety, and depression are coexistent events, and they have intermingled biological processes and manifestations. Therefore, they are often studied together in the gut microbiota–brain axis domain [42]. These most prevalent psychiatric conditions involve the failure of allostasis, which is a psychological stress response that restores homeostasis in the body. The dynamic modulation of the body’s stress response systems, involving neuroendocrine signaling through the HPA axis, which regulates BDNF and glucocorticoid production, is vital in learning and memory formation [40].

Correspondingly, probiotic treatment has been considered a potential therapeutic approach for addressing behavioral deficits via inducing changes in the gut microbiota and BDNF expression [43]. Another study showed that specific-pathogen-free mice treated with oral antimicrobials showed transiently perturbed gut microbiota, leading to increased hippocampal BDNF concentrations and the subsequent prevention of depression [44]. Additionally, two weeks of combined treatment with *Lactobacillus helveticus* R0052 and *Bifidobacterium longum* R0175 resulted in increased hippocampal expression of BDNF in stressed mice, thus reversing synaptic dysfunction and eventually improving learning and memory task performance [45]. Hence, the gut microbiota has a promising ability to control BDNF and regulate the development of depression-like behavior.

Patients with mental conditions (depression) exhibit gut microbiome dysbiosis [46]. Another study established that the gut microbiome dysbiosis in patients with depression is associated with decreased BDNF levels, whereas *Faecalibacterium* is associated with clinician depression score bars, which indicate the severity of depressive symptoms [47]. Multiple factors transmit signals to the brain regarding the gut state, such as infectious constituents, antibiotics, vagal sensory fibers, and cytokines. The HPA axis regulates metabolism, microbiome diversity, and nutrient absorption [48].

Few studies have provided evidence that the administration of bacteria with psychobiotic properties for a fixed period results in alterations in anxiety and depression scores. The administration of *Bifidobacterium infantis* showed similar effectiveness as citalopram (an antidepressant drug) [49]. Similar results were obtained with the administration of another psychobiotic bacterium, *Lactobacillus rhamnosus* [50].

An exciting result was obtained in an experiment in which feces from patients with depression were transferred to microbiota-depleted rats. Fecal microbiota transplantation induced some depression-related features and associated physiological changes in recipient animals, suggesting that the gut microbiota might play a causal role in the development of depression and provide a tractable target for the treatment and prevention of this disorder [51].

Previous reports indicate that patients with depression have mild chronic inflammation [52,53]. In this inflammation–depression association, speculation continues regarding which component is the cause and which is the effect. Studies from the last decade have demonstrated that systemic inflammation and neuroinflammation result in depression and leaky gut [47,54,55]. The microbiome may yield a new class of psychobiotics for the treatment of anxiety, depression, and other mood disorders [56].

Studies have suggested that inflammation of the GIT results in neuroinflammation, which fuels microglial action and initiates the kynurenine pathway. These processes lead to depression [57].

#### 2.3.2. Role of BDNF in Irritable Bowel Syndrome Induced Depression

The physiopathology of irritable bowel syndrome (IBS) involves the multifactorial and bidirectional dysfunction of the brain–gut axis, including epigenetics, proinflammatory cytokines, visceral hypersensitivity, altered neuropeptides, and GI motility. IBS is associated with a high risk of comorbid depression, and recent studies have evaluated the role of BDNF in IBS. Elevated mucosal BDNF may be involved in the pathogenesis of IBS by elevating mucosal nerve growth and visceral sensitivity [58]. A comprehensive study revealed that BDNF is detectable in the human descending colon, with BDNF mRNA levels being higher in female IBS patients than in male patients, supporting sex hormone involvement in the regulation of BDNF [59]. Additionally, a ketogenic diet has been shown to produce a dual effect in improving the 5-HT and BDNF systems. The rats exhibited an upregulation of the BDNF receptor TrkB as a counteracting response to a stress-induced reduction in neurotrophin levels [60].

#### 2.3.3. Findings Related to Gut Microbiota in Patients with Depression

Many clinical studies have reported that the gut microbiota composition in patients with depression is abnormal compared with healthy controls, further suggesting the vital role of the brain–gut–microbiota axis in depression [47,61,62,63,64,65]. A recent report revealed that gut microbiota DNA contributes to the depression phenotype in patients with distinctive depression compared to that in healthy controls [66]. A detailed study including patients with depression adopted a multilevel omics viewpoint to understand changes in the gut microbiota and metabolomics to explore gut ecology in the etiology of depression [65]. The overall phenotype of gut ecology, bacteriophages, fecal metabolites, and bacterial species significantly changed in patients with depression compared to that in healthy controls. Different metabolites, including those of *Oscillibacter* sp. *ER4*, *Blautia wexlerae*, and *Blautia* sp. *Marseille-P2398* and ʟ-homoserine phosphate, in the altered gut may be vital markers of depression [65]. A study demonstrated a distinct difference in the gut microbiota between individuals with depression and healthy controls, where the group with depression showed major alterations associated with the *Bacteroidaceae* family [67]. Many systematic reviews and meta-analyses have been conducted to assess evidence of gut microbiota perturbations in patients with depression. A meta-analysis showed that the genera *Corprococcus* and *Faecalibacterium* were reduced in patients with depression compared to controls and that depressive symptoms improved in interventional studies with probiotics [68]. Another recent meta-analysis showed that a disturbance in the gut microbiota was connected to a transdiagnostic pattern with a reduction in some anti-inflammatory butyrate-producing bacteria and an enhancement of proinflammatory bacteria in patients with psychiatric conditions, including depression [69].

Recent clinical studies have shown that depression is linked with the gut microbiota and brain separately, and few studies have focused on bidirectional communication; however, more clinical studies need to be performed in the future for a better understanding of the gut–brain axis with respect to depression, and the direction of causality between the two entities needs to be determined.

### 2.4. Role of BDNF-mTORC1 Signaling Pathway in Depression

The mammalian target of rapamycin complex-1 (mTORC1) is a major growth regulator [70]. BDNF affects the nervous system through the BDNF–mTORC1 pathway. Numerous clinical studies have shown that ketamine and scopolamine improve synapse maturity via the activation of the BDNF–mTOR pathway, which upregulates the expression of several synapse-related proteins; in contrast, blocking mTOR signals can fully disturb the behavioral response of these synapses [71,72]. This may represent a unique and fast-acting antidepressant mechanism. Some reports have established that hypidone hydrochloride activates neurons by reversing the inhibitory effect of the serotonin receptors on the GABAergic neurons that follow the BDNF–mTORC1 pathway, producing an antidepressant effect [73,74,75,76,77]. The fast-acting antidepressant effect of ketamine and its active metabolite (2R,6R)-hydroxyketamine was blocked by a BDNF-function-blocking antibody or rapamycin [78,79], a classical inhibitor of mTORC1 [80]. A potent antidepressant (NV-5138, an mTORC1 activator) elevated the synaptic proteins by upregulating mTORC1 signaling, in which BDNF is required to participate [81]. These findings suggest that mTORC1 is a promising therapeutic agent for the development of antidepressants.

In summary, the latent mechanism of the fast antidepressant action of the BDNF-mTORC1 signaling pathway involves, first, the inhibition of the activity of GABA interneurons by the release of glutamate from glutaminergic neurons. Second, the activation of AMPA receptors induces BNDF release via the stimulation of L-type voltage-dependent calcium channels. Finally, this BDNF triggers TrkB, Akt, ERK, and AMPK and subsequently initiates the mTORC1 pathway, which results in the increased production of proteins that are involved in synaptic formation (e.g., GluA1 and PSD95) and further enhances the frequency and amplitude of the excitatory postsynaptic current, thus promoting the growth of neurons and synapses and producing an antidepressant effect [82,83]. 

### 2.5. Effects of Antidepressants on BDNF and Their Related Side Effects

BDNF and 5-HT control neurogenesis, synaptic plasticity, and neuronal survival in the brain. They regulate each other such that 5-HT stimulates the expression of BDNF, which boosts the growth and survival of 5-HT neurons. An imbalance in 5-HT and BDNF signaling results in depression and anxiety disorders [84]. The 5-HT–BDNF interaction mediates the therapeutic efficacy of antidepressant agents through neuroplastic and epigenetic mechanisms [85].

Presently, it is unclear how antidepressant drugs bind to their targets to induce antidepressant effects. The BDNF receptor promotes neuronal plasticity and antidepressant responses. A recent study discovered that binding to TrkB and the allosteric facilitation of BDNF signaling are common mechanisms for antidepressant action. For example, BDNF binding to TrkB may change neuronal excitability and regulate synapse formation. As a result, the downregulation of BDNF may induce depression. Therefore, a significant positive correlation can exist between BDNF levels and depression. For example, classical and rapid-action antidepressants directly bind to TrkB, thus enabling its synaptic localization and its activation by BDNF [86].

### 2.6. Modulation of BDNF Signaling by Different Natural Products and Their Derivatives

Natural products are sourced from a wide variety of potential derivatives belonging to different families and classes that produce pharmacological effects in the gut–brain axis, targeting BDNF to exhibit antidepressant effects via different signaling pathways and actions. 

#### 2.6.1. Flavonoids, Flavanones, and Flavanol

##### Apigenin

Apigenin (4′,5,7-trihydroxyflavone) is the most common flavonoid, with a biological activity that is widely present in many plants. It produces multiple pharmacological effects, including antioxidant, antitumor, anti-inflammatory, neuroprotective, and nephroprotective effects. It is considered a major ingredient of *Petroselinum crispum,* commonly known as parsley, a common vegetable. Several clinical trials have shown that apigenin is a potent antidepressant. Apigenin has been reported to ameliorate learning and memory impairment by relieving the Aβ burden, suppressing the amyloidogenic process, inhibiting oxidative stress, and restoring the ERK/CREB/BDNF pathway [87]. The antidepressant effects of apigenin indicate that the antidepressant-like mechanism of apigenin is mediated by the upregulation of BDNF levels in the hippocampus [88]. A recent study showed that apigenin treatment prevents cognitive deficits and recovers behavioral impairments without changing seizure severity in kindled mice, which can be attributed to CREB–BDNF upregulation in the hippocampus [78].

##### Baicalein

Baicalein is a trihydroxyflavone and a bioactive flavonoid that was originally isolated from the roots of the traditional Chinese herbs *Scutellaria baicalensis* and *Scutellaria lateriflora*. Previous studies have indicated that baicalein could serve as a promising therapeutic agent for neurodegenerative diseases [79]. A recent study revealed that the administration of flavonoids extracted from *Scutellaria baicalensis* Georgi significantly inhibited neuroinflammation and maintained neurotransmitter homeostasis. It significantly improved Parkinson’s-disease-related depression and activated the BDNF/TrkB/CREB pathway [89]. Baicalein has been shown to attenuate LPS-induced depression-like behavior by suppressing neuroinflammation and inflammation induced by peripheral immune responses. It additionally elevated mBDNF levels in the hippocampus of LPS-treated mice and increased the mBDNF/proBDNF ratio, which regulates neuronal survival and synaptic plasticity [90].

##### Hesperidin

Hesperidin (4′-methoxy-7-*O*-rutinosyl-3′,5-dihydroxyflavanone) is a naturally occurring flavanone glycoside found in foods, including grapefruits, oranges, and lemons, and it has multiple pharmacological activities. Interestingly, hesperidin treatment significantly reduced depression-related symptoms in mice with mild traumatic brain injury. This antidepressant-like effect of hesperidin is mediated by decreased neuroinflammation and oxidative damage and enhanced BDNF production in the hippocampus [91]. Hesperidin efficiently exerted neuroprotective effects in depression and reduced chronic unpredictable mild stress (CUMS) in depressed mice via the NF-κB and BDNF/TrkB pathways [92]. Other recent studies have described a role of hesperidin in reducing depression via corresponding mechanisms based on the NLRP3 inflammatory signaling pathway [93].

##### Luteolin-7-*O*-glucuronide

Luteolin-7-*O*-glucuronide (L7Gn) is a flavone glycoside found in various plants, including *Perilla frutescens*, *Remirea maritima*, *Codariocalyx motorius*, and *Ixeris dentata*. A recent study illustrated that L7Gn has therapeutic potential for sleep-deprivation-induced stress via the activation of BDNF signaling. Additionally, L7Gn increased BDNF mRNA and protein levels, which were reduced by SD stress, and an L7Gn intervention resulted in the upregulation of TrkB, ERK, and CREB, which are downstream molecules of BDNF signaling, reducing depression in rat and mouse models of inflammation [94].

##### Naringenin

Naringenin (4’,5,7-trihydroxyflavanone) is a dietary flavonoid that is abundant in citrus fruits (mandarins, grapefruit, and lemons) and vegetables. It has been reported to produce multiple biological effects on obesity, hypertension, cardiovascular diseases, metabolic syndrome, and neurodegenerative disorders [95]. Naringenin produces an antidepressant-like effect mediated by the activation of BDNF signaling in the hippocampus [96]. Naringenin has been shown to improve depressive- and anxiety-like behaviors in mice exposed to repeated hypoxic stress by modulating oxido-inflammatory mediators and NF-κB/BDNF expression [97].

##### Quercetin

Quercetin, a naturally occurring flavonoid derived from many plants and foods, has been reported to show neuroprotective and antidepressant effects; however, its underlying mechanisms are unclear.

Quercetin alleviated LPS-induced depression-like behavior and impaired learning in rats caused by a BDNF-related imbalanced expression in the hippocampus and PFC [98]. A recent study revealed that quercetin exerts an antidepressant-like effect through the upregulation of the CREB/BDNF signaling pathways [99]. Recent reports describe that quercetin significantly decreases antibiotic-linked gut dysbiosis and in return avoids gut dysbiosis cognitive dysfunction in mice [100].

##### Sanggenon G

Sanggenon G is a novel, natural, nonpeptidic, active constituent of *Morus alba*, and it has been used as an important medicinal agent for stress since ancient times. Intervention with the root bark of *Morus alba* (RBM) in diabetic rats upregulated BDNF expression and the phosphorylation of ERK and Akt in the prefrontal cortex. The results suggested that RBM could minimize the depression-like behaviors induced by diabetes, suggesting the therapeutic potential of RBM for diabetes-associated depression [101].

##### Silibinin

Silibinin, a flavonoid derived from the herb milk thistle (*Silybum marianum*), has been used as a hepatoprotectant for the clinical treatment of liver disease. Silibinin might serve as a potential therapeutic drug for neurodegenerative diseases such as depression, anxiety, and memory loss because several studies have been conducted to determine its antidepressant effect along with its mechanism of action. A study reported that silibinin ameliorated the impairment of learning and memory in LPS-injected rats via the activation of the ROS-BDNF-TrkB pathway in the hippocampus [102]. Similarly, another study suggested that silibinin exhibits antidepressant effects through the BDNF/TrkB signaling pathway, improving neural stem cell proliferation in acute depression [103,104].

##### Orange Juice

Numerous studies have focused on the intake of flavonoids through vegetables and fruits to reduce the risk of depression [105,106]. A randomized controlled study was conducted among young Korean adults in treatment and placebo groups. The results provided novel interventional evidence that an alteration in the microbiome due to flavonoid treatment is related to a potential improvement in depression in young adults. Notably, the abundance of the Lachnospiraceae family (Lachnospiraceae_uc, Eubacterium_g4, Roseburia_uc, Coprococcus_g2_uc, and Agathobacter_uc) increased in the group treated with flavonoid-rich orange juice compared to that in the placebo group. Positive correlations have been observed between BDNF and the Lachnospiraceae family [107].

#### 2.6.2. Phenol, Polyphenol, and Phenolic Acid Derivatives

##### Chlorogenic Acid

Chlorogenic acid is a traditional Chinese medicine that is abundantly present in honeysuckle and Eucommia; it exhibits diverse biological and pharmacological effects. Syringaresinol–di–*O*–β-ᴅ-glucoside is also called eleutheroside E or isofraxidin. Many studies have demonstrated the potent antidepressant activity of chlorogenic acid through different pathways; however, a recent study found that the combined administration of chlorogenic acid and (+)-syringaresinol–di–*O*–β-ᴅ-glucoside significantly increased hippocampal BDNF protein expression and modulated autonomic regulation, which improved anxiolytic behavior [108]. More recently, chlorogenic acid has been reported to improve high-fructose and high-fat diet-induced cognitive damage by targeting the microbiota–gut–brain axis. It promotes phosphorylcholine while reducing the level of energy metabolism substrates, which decreases the levels of neuroinflammatory factors, including IL-1β, TNF-α, and NF-Kβ, in experimental animal models [109].

##### Curcumin

Curcumin, a phytochemical derived from the rhizome of *Curcuma longa*, displays a wide range of pharmacological activities, including potent anti-inflammatory, anticarcinogenic, neuroprotective, and antioxidative activities. Curcumin potentially improves depression-like behavior by modulating stress hormones, hippocampal neurotransmitters, and BDNF levels in rats [110]. It exhibits an antidepressant effect associated with an increase in hippocampal BDNF and ERK [111,112], along with several other signaling pathways targeting the gut–brain axis.

##### Ellagic Acid

Ellagic acid (EA; 2,3,7,8-tetrahydroxybenzopyrano [5,4,3-cde]benzopyran-5-10-dione), a natural polyphenolic compound, is present in natural plant-based foods such as raspberries, strawberries, grapes, pomegranates, and walnuts. In the past decade, EA has attracted the attention of scientists owing to its large array of pharmacological properties, including antioxidant, anti-inflammatory, neuroprotective, antidepressant, antianxiety, and antinociceptive properties [113]. A study demonstrated that EA exerted an antidepressant-like effect in depressed mice, which was mediated by increased BDNF levels in the hippocampus of mice [114]. EA can be used as a potential nutraceutical for the treatment and management of depression. It attenuates CUMS-induced hippocampal damage, significantly enhances BDNF and serotonin levels, and suppresses the inflammatory response [115].

##### Epigallocatechin-3-gallate

Epigallocatechin-3-gallate (EGCG) is the major catechin and the most active and abundant polyphenol in green tea. Tea is the second most consumed beverage in the world, after water. (−) Epicatechin flavanols are found in natural products such as cocoa and green tea. They reduces anxiety by increasing hippocampal BDNF expression, along with elevated levels of pAkt in the hippocampus and cortex [116]. EGCG is a potent nutritional supplement that reduces the cognitive dysfunction associated with postmenopausal depression. A recent study found that EGCG simultaneously improved cognitive impairment by rescuing long-term synaptic plasticity. This occurs through the restoration of silent synapse formation by increasing hippocampal BDNF-TrkB [117]. More recent studies have indicated that in the hippocampus EGCG provides relief from depression by downregulating IL-1β and upregulating BDNF in a mouse model.

A jasmine tea intervention significantly attenuated CUMS-induced depression-like behavior in rats by upregulating BDNF and 5-HT expression in the hippocampus and cerebral cortex. It regulates the composition of the gut microbiota (*Patescibacteria*, *Bacteroidetes*, *Spirochaetes*, *Elusimicrobia*, and *Proteobacteria*) through metabolic pathways [118].

##### Eleutheroside E

*Acanthopanax senticosus* Harm (ASH), also known as Siberian ginseng or eleuthero, is commonly found in China, the Republic of Korea, Russia, and Japan. Eleutherosides are the most abundant active constituent of ASH. ASH has many pharmacological properties that enable its application as a therapeutic tool for various diseases such as hypertension, cancer, rheumatoid arthritis, allergies, chronic bronchitis, diabetes, and neurodegenerative diseases. An experiment found that ASH treatment significantly elevated hippocampal BDNF protein expression and suggested that the antidepressant effects of ASH occur through the regulation of autonomic function and increased hippocampal BDNF signaling [119]. A recent study indicated that supplementation with eleutheroside E disturbs the microbiota, activating the PKA/CREB/BDNF signaling pathway, which in turn improves cognitive impairment in irradiated mice [120].

##### Eugenol

Eugenol, a pale yellow liquid extracted from the essential oil of mustard leaves (*Brassica juncea* var. *crispifolia* L. H. *Bailey*), belongs to the mustard family (Brassicaceae or Cruciferae), which has many common names, such as brown mustard, Indian mustard, Chinese mustard, or oriental mustard. Eugenol has been shown to promote metallothionein-III production in the hippocampus and exhibit antidepressant-like effects in mice. Mustard leaves contain numerous phytochemicals, such as chlorophyll, β-carotene, ascorbic acid, potassium, calcium, and other minerals [121]. These compounds exert therapeutic effects. A study suggested that mustard leaf extract is an agonist that helps reduce depression and stress. They reduce stress by regulating hormones and neurotransmitters in CRS mice as well as BDNF expression and apoptosis in the brain [122]. Recent studies have shown that mice fed a high-fat diet with eugenol exhibit reduced adiposity and modified gut microbiota [123].

##### Resveratrol

Resveratrol (3,5,40 -trihydroxystilbene) is a polyphenol predominantly present in *Polygonum cuspidatum*, the skin of red wine, red grapes, and some nuts. It exhibits a potent therapeutic action against depression by upregulating BDNF levels in the hippocampus. A previous study reported that the administration of resveratrol ameliorated depressive behavior in CUMS rats by reducing HPA axis hyperactivity and enhancing BDNF levels. [124]. The gut–brain axis is improved by the regulation of BDNF-dependent signaling in rat models of IBS. Resveratrol exhibited anti-IBS-like effects on depression, stress, and intestinal motility abnormalities by regulating 5-HT-dependent PKA–CREB–BDNF signaling in the brain–gut axis [125].

##### Thymol

Thymol (2-isopropyl-5-methylphenol) is a natural monoterpenoid phenol found in high concentrations in the oils of thyme from different *Thymus* species such as *T. vulgaris*, *T. spicata*, and *T. ciliates* and *Nigella sativa* seeds [126]. Several pharmacological activities of thymol have been defined, such as fungicidal, antibiotic, and anti-inflammatory activities. Some studies have also found that thymol reduces depressive symptoms (via the upregulation of BDNF levels), restores anhedonia and short-term memory, and improves anxiety [127].

##### 2,3,5,4′-Tetrahydroxystilbene-2-*O*-β-d-glucoside

2,3,5,4′-Tetrahydroxystilbene-2-*O*-β-d-glucoside (THSG) is the major bioactive compound of a traditional Chinese herb (*Polygonum multiflorum*) that has been used for years for black hair coloring and tonifying the liver and kidneys. Pharmacological studies have verified the antiaging, antioxidant, anti-inflammatory, and neuroprotective effects of THSG. Additionally, THSG is effective in decreasing stress-induced depression by ameliorating neurotrophins and their related signaling pathways. Therefore, exploring THSG in folk medicine might enable the identification of BDNF-targeting agents for antidepression treatment [128].

#### 2.6.3. Fatty Acids, Lignan, Steroids, and Terpenoids

##### Ginsenoside Rd

Ginsenoside Rd, one of the most active constituents of *Panax ginseng* berry, has been reported to have pharmacological effects on the CNS, including protection against neurotoxicity. In an experimental study in mice, Rd exhibited an antidepressant-like effect mediated by the hippocampal BDNF signaling pathway [129]. Bifidobacteria-fermented red ginseng (fRG) alleviated *Escherichia coli* (EC)-induced gut dysbiosis by increasing the Bacteroidetes population and reducing the Proteobacteria population. Additionally, red ginseng and its constituent ginsenosides alleviated EC-induced anxiety/depression and colitis. In summary, fRG and its constituents mitigated anxiety/depression and colitis by regulating NF-κB-mediated BDNF expression and gut dysbiosis in rodents [130].

##### Limonene

Limonene is a cyclic terpene found in citrus fruits such as *Citrus sinensis*, a sweet orange belonging to the Rutaceae family. One study evaluated the effect of orange essential oil (OEO) and its main component (limonene) in depressive mice. The results indicated that OEO inhalation significantly ameliorated the depression-like behavior of CUMS mice, with decreased body weight, sucrose preference, curiosity, and mobility. Improvements in the neuroendocrine, neurotrophic, and monoaminergic systems are related to the antidepressant effects of limonene in mice [131]. A recent study demonstrated that *C. reticulata* essential oils (CREOs), which contain *d*-limonene as the main component, can serve as a potential therapeutic drug and food supplement against depression [132].

##### Omega-3 Fatty Acids

Omega-3 fatty acids (alpha-linolenic, eicosapentaenoic, and docosahexaenoic acids) are present in most Western diets and in vegetable oils, nuts (especially walnuts), flax seeds, and fish oil. These essential fats play a critical role in CNS functions. Investigations have linked omega-3 fatty acids to various neuropsychiatric disorders, including depression and anxiety. They exhibit therapeutic effects against different diseases, particularly mood-related disorders and memory improvement. A novel mechanism involving CREB and BDNF is the modulation of omega-3 [133]. Omega-3-enriched dietary supplementation offers protection against reduced plasticity and impaired learning ability. It normalizes the levels of BDNF and associated synapsin I and CREB, reduces oxidative damage, and counteracts learning disability [134].

##### Oleanolic Acid

Oleanolic acid (3β-hydroxyolean-12-en-28-oic acid), a natural pentacyclic triterpenoid, is widely present in food and medicinal plants such as *Olea europaea* (olive oil), raisins, and dried cranberries. Oleanolic acid is a potential therapeutic agent for the treatment of cognitive deficits and is primarily a mediator of the functional impairment caused by major depressive disorder. Oleanolic acid improves memory impairment by modulating the BDNF–ERK1/2–CREB pathway through TrkB activation in mice [135]. Another recent study reported that oleanolic acid changes the gut microbiota and immune-related gene expression in intestinal epithelial cells [136].

##### *Poria* *cocos*

*Poria cocos* is an edible medicinal mushroom from the Polyporaceae family that has been used in China for 2000 years as a complementary therapy for managing depression and anxiety [137]. Triterpenoids are the most active compounds of *Poria cocos,* and their intervention significantly improved depression-like behavior in CUMS rats and restored the BDNF levels and neural growth in the hippocampus, which was impaired by depression. They presented a potential antidepressant and antianxiety effect mediated by the intestinal microbiota and cecal content metabolites. The triterpenoids of *Poria cocos* markedly upregulated the relative population of healthy microbes. The gut microbiota is strongly associated with cecal content metabolites, especially compounds related to energy metabolism, inflammation, and immunity [138]. In APP/PS1 mice, *Poria cocos* improved cognitive function by restoring the balance between the production and removal of Aβ and reversing the imbalance in the microbial community. A recent study revealed that it upregulates brain–gut peptides and improves the gastrointestinal mucosa and immunity, improving the function of dyspepsia. The reason behind its potential pharmacological effect is the presence of terpenoids and polysaccharides that regulate the imbalance in the microbial community to ameliorate depression and anxiety via the TNF-α/NF-κβ signaling pathway [139].

##### Sesamol

Sesamol (SML, 3,4-methylenedioxyphenol), a natural lignan present in the extract of *Sesamum indicum* L., has several bioactivities, including antioxidative, lipid-lowering, and anti-inflammatory effects [140]. A study illustrated that the alcoholic extract of sesame (*Sesamum indicum* L.) cake (SLE) and sesamol markedly improved CUMS-induced depression and memory loss by increasing BDNF and PSD-95 (postsynaptic density protein 95) in depressed mouse brains [141].

Another study was conducted to determine the effect of sesamol in treating depression and anxiety by targeting the gut–brain axis. The results illustrate that sesamol is a novel nutritional intervention strategy for preventing IBD and its symptoms of anxiety and depression. As one of its many effects, sesamol has been shown to improve BDNF levels by upregulating the BDNF/TrkB/CREB signaling pathway [142].

#### 2.6.4. Anthraquinone and Anthocyanin

##### Cyandin-3-glucoside and Cyanidin-3-sophoroside-5-glucoside

Anthocyanins, a widespread class of flavonoids in fruits and vegetables, are conventionally present in the daily diet (50 mg/day); they exhibit strong free radical scavenging and anti-inflammatory properties. Purple cauliflower is a rich source of anthocyanins with high antioxidant activity. A recent study proposed that the anthocyanin extract from blueberries ameliorates depression-like behavior in CUMS by upregulating monoamine neurotransmitter levels and BDNF expression and inhibiting MAOA, which promotes neurogenesis via the ERK/CREB/BDNF signaling pathways [143]. Cyandin-3-glucoside and cyanidin-3-sophoroside-5-glucoside are the major anthocyanins. However, it needs to be determined whether the antidepressant effect of cauliflower is due to the presence of these compounds in the gut–brain axis experimental model.

##### Hypericin

*Hypericum triquetrifolium*, commonly called curled-leaved St. John’s wort, is a plant species of the Hypericaceae family. It is an herbaceous perennial plant and a native Iranian species found in the southwestern part of the country. *H. triquetrifolium* extract has been shown to reduce stress via the BDNF signaling pathway; a study demonstrated that treatment with *H. triquetrifolium* could ameliorate stress-associated hippocampus-dependent memory deficits through a mechanism that might involve BDNF in the hippocampus [144]. According to the literature, *H. triquetrifolium* contains hypericin (one of the most active compounds), flavonoids, phenolic compounds such as chlorogenic acid, and essential oils [145]. Hypericin can effectively relieve postpartum depression by reversing glucocorticoid metabolism and relieving neuroinflammation [146].

#### 2.6.5. Carotenoids

##### Fucoxanthin

Fucoxanthin is a natural orange carotenoid and a typical lipid compound found in edible brown algae and seaweeds. Fucoxanthin belongs to the class of non-pro-vitamin A carotenoids and presents several health benefits, including antioxidant, apoptosis-promoting [147], anticancer, anti-inflammatory, and antidiabetic effects [148]. Fucoxanthin plays a crucial role in combating neurodegenerative diseases by decreasing proinflammatory factor production or increasing the neuroprotective expression of BDNF. It mediates reactive oxygen species (ROS) production by activating the PKA/cAMP and CREB pathways and promotes BDNF activity [149]. In addition, through the AMPK-NF-κB pathway, fucoxanthin protects mice from lipopolysaccharide-induced depression-like behavior [150].

##### Lycopene

Lycopene is a lipid-soluble pigment of carotenoids that is found in various fruits and vegetables, including watermelon (*Citrullus lanatus*) and tomatoes (*Solanum lycopersicum*); it can distribute in the brain and intestine and cross the blood–brain barrier [151]. It improves post-traumatic stress disorder (PTSD)-like behavior in mice by rebalancing neuroinflammatory and oxidative stress in the brain. The restoration of BDNF expression may be a potential mechanism underlying the anti-PTSD effects of lycopene [105,152]. Lycopene has been shown to improve dextran sulfate sodium induced colitis depression and behavioral disorders via balancing the microbiota–gut–brain axis balance by upregulating BDNF expression and postsynaptic density protein. It reshaped the gut microbiome in colitis mice by decreasing the relative abundance of Proteobacteria and increasing the relative abundance of Bifidobacterium and Lactobacillus [153].

#### 2.6.6. Bodily Fluids

##### Nicotinamide Riboside

Nicotinamide riboside (NR), a form of vitamin B_3_, is a nicotinamide adenine dinucleotide (NAD^+^) precursor that is mainly obtained from cow milk and yeast. Recent studies have reported that NR has neuroprotective effects. A previous study showed that NR intervention significantly upregulated BDNF and reduced the inhibition of the AKT signaling pathway in the hippocampus of treated mice. Therefore, dietary supplementation with NR diminished alcohol-induced depression-like behaviors by altering the composition of the gut microbiota [143].

##### Honey

Tualang honey is produced by the giant honey bee (*Apis dorsata*) in South and Southeast Asia and is reported to exhibit strong antioxidant properties. It contains various phytochemicals, such as flavonoids, phenolic acids, glucose oxidase, carotenoid derivatives, and organic acids. Tualang honey reduced depression-like behavior in stressed ovariectomized rats via restoring the HPA and increasing BDNF levels [154]. Another study revealed that Tualang honey protects against memory decline due to stress exposure or aging by increasing PFC and maintaining the hippocampal morphology, possibly by ameliorating oxidative stress in the brain or upregulating BDNF levels [155]. Recent studies have indicated that the improvement in cognitive function is a combined effect of the exceptional metabolites of honey, which may be different for different types of honey. The microbial community in the gut stimulates sucrose sensitivity and alters neurotransmitter signaling in the brain, which reverses cognitive dysfunction via the upregulation of BDNF.

#### 2.6.7. Medicinal Plants and Herbal Formulations

##### *Cinnamomum* *zeylanicum*

Cinnamon (*Cinnamomum zeylanicum)* is one of the most famous and oldest medicinal plants worldwide. This evergreen tree, including almost all parts, is considered a remedy for respiratory, digestive, and gynecological diseases [156]. A recent study claimed that the hydroalcoholic extract of *Cinnamomum* (HEC) ameliorated depression symptoms in rats through the upregulation of the BDNF protein and its TrkB receptor in the prefrontal cortex [157]. Additionally, a subcritical water extract of *Cinnamonum japonicum* suppressed dextran sodium sulfate induced intestinal damage in a mouse colitis model.

##### *Dendrobium* *officinale*

*Dendrobium officinale* is a precious plant used in medicine and food products that contains several bioactive components, such as polysaccharides, bibenzyls, phenanthrenes, and flavonoids. This plant is recorded in the Chinese Pharmacopoeia, which is widely used in traditional Chinese medicine [158].

A recent study demonstrated that the alcohol-soluble polysaccharides of *D. officinale* exhibited a significant antidepressant effect by modulating gut bacterial and fungal homeostasis and the levels of short-chain fatty acids to reduce intestinal barrier disruption and excessive inflammatory responses; thus, protective effects were achieved by neuronal apoptosis and the maintenance of the serotonin system through the activation of the BDNF–TrkB–CREB pathway [159].

##### Huanglian-Jie-Du-Tang

Huanglian-Jie-Du-Tang (HJDT) is an herbal formula composed of traditional Chinese roots such as *Coptis chinensis* Franch., *Phellodendron chinense* Schneid., *Scutellaria baicalensis* Georgi, and *Gardenia jasminoides* Ellis, which are easily found in China, Japan, and the Republic of Korea. It exhibits many health benefits and helps combat diseases such as GIT-associated conditions, Alzheimer’s disease, and ischemic brain injury. A study in rats with chronic unpredictable stress revealed that HJDT might be a latent Chinese medicine used for treating or alleviating complex symptoms of depression via the BDNF–TrkB–CREB pathway [160]. Studies using various integrated omics approaches showed that the Huanglian Jiedu decoction altered the peripheral microenvironment to slow the course of Alzheimer’s disease via the gut–brain axis.

##### *Cocos nucifera* Husk Fiber

The coconut (*Cocos nucifera*) is a tropical fruit tree belonging to the family Arecacecae, and it has been used in several food products and beverages. Scientific evidence suggests medicinal and health-related benefits [161] for different parts of this plant, such as coconut water and husks. The husk of *Cocos nucifera* has been shown to produce an antidepressant effect in mice by elevating the BDNF levels in the hippocampus. Therefore, their use could serve as a novel strategy in the development of antidepressant drugs [162].

##### *Semen Sojae* Praeparatum

*Semen Sojae* Praeparatum is a traditional Chinese medicine that is naturally fermented from soybean, mulberry leaf, and sweet wormwood herbs and is officially listed in the Chinese Pharmacopoeia. A recent experiment indicated that *Semen Sojae* praeparatum is fermented by *Rhizopus chinensis* 12 and *Bacillus* sp. DU-106 improved the neurotransmitter levels and morphology of hippocampal neurons associated with the modulation of the gut microbiota in rats with mild stress. It upregulated the levels of 5-HT, BDNF, and GABA in the hippocampus. Furthermore, *Semen Sojae* Praeparatum is a novel therapeutic agent against depression, according to a theoretical investigation of the relationship between the microbiota–gut–brain axis and antidepressants [163]. A recent study revealed that a Zhi-Zhi-Chi decoction consisting of *Semen Sojae* Praeparatum and *Gardenia fructus* shows an antidepressant effect by altering the gut microbiota to promote butyrate synthesis, which in turn regulates BDNF, neurotransmitters, endocrine, and anti-inflammatory responses along the gut–brain axis.

#### 2.6.8. Phthalide

##### 3-n-Butylphthalide

3-n-Butylphthalide (NBP; (±)-3-butyl-1(3H)-isobenzofuranone), a family of compounds initially extracted from *Apium graveolens* Linn. (seeds of Chinese celery) are responsible for the aroma and taste of celery. This is a new drug approved by the State Food and Drug Administration (SFDA) in 2002 for the treatment of ischemic stroke [164]. Current studies have revealed that NBP exhibits several pharmacological roles, such as neuroprotective, antioxidant, antiapoptotic, and platelet aggregation inhibition roles. One study reported that NBP acts as a therapeutic agent for depression-like behaviors through the modulation of the serotonergic system and BDNF–ERK–mTOR signaling [165]. In recent studies, NBP has been used as a positive control for evaluating the therapeutic effect of a Buyang Huanwu decoction in cerebral ischemia via the targeting of the gut microbiota.

#### 2.6.9. Probiotics

##### *Clostridium butyricum* (*C. butyricum*)

*C. butyricum*, a Gram-positive, spore-forming, obligate anaerobic rod bacterium, is found in the feces of 10–20% of healthy humans. It is an important intestinal probiotic that plays a vital role in regulating the gut microbiota [166]. *C. butyricum* has a prominent antidepressant effect, and it significantly improved CUMS-induced depression-like behavior in mice by increasing 5-HT, GLP-1 (glucagon-like peptide-1), and BDNF expression [167]. Another study indicated that mice exposed to prolonged social defeat stress exhibited depression-like behavior that was prevented by *Clostridium butyricum miyairi* 588 (CBM588), which also controls microglial activation.

##### *Porphyromonas* *gingivalis*

*Porphyromonas gingivalis* (Pg) downregulates BDNF maturation, leading to depression-like behavior in mice. Therefore, a study suggested that Pg is a modifiable risk factor for depression and uncovered a novel therapeutic target for the treatment of depression [168].

##### *Bifidobacterium longum* NCC3001

*Bifidobacterium longum* NCC3001 has been shown to produce an anxiolytic effect via vagal pathways for gut–brain communication without altering BDNF levels. Additionally, it has been reported to reduce the excitability of enteric neurons, which might send signals to the central nervous system by activating vagal pathways at the level of the enteric nervous system [169]. In addition, a recent pilot study found that the probiotic administration of *B. longum* NCC3001 significantly improved depression in a cohort of individuals with IBS by upregulating BDNF [170].

##### *Lactobacillus casei* (*L. casei*)

*L. casei* is a probiotic associated with the modification of intestinal microbiota, and the gut microbiota–brain axis has been demonstrated to play an important role in many CNS diseases.

*L. casei* administration improved CUMS-induced depression-like behavior and amended the gut microbiota structure, which was impaired by CUMS, in rats. Moreover, an *L. casei* intervention overturned CUMS-induced changes in the protein expression of BDNF and its receptor TrkB as well as the CUMS-induced activation of the ERK1/2 and p38 MAPK signaling pathways. This intervention study demonstrated that *L. casei* could significantly protect against depression in rats, which was possibly associated with alterations in the gut microbiota composition and the mediation of BDNF–TrkB signaling [171]. *L. casei* exhibited antidepressant potential in a rat model of postpartum depression, mediated by the microbiota–gut–brain axis. An experimental study suggested that *L. casei* could improve postpartum depression by altering the composition of the gut microbiota, brain monoamines, and oxidative stress, which might be related to the regulation of the BDNF–ERK1/2 pathway [172].

##### *Lactobacillus helveticus* R0052

Psychobiotics have been defined as probiotics upon ingestion in amounts adequate to produce a positive influence on mental health, which is attributed to their impact on the microbiota–gut–brain axis [173]. The current psychobiotic formulation (*Lactobacillus helveticus* R0052 and *Bifidobacterium longum* R0175) treatment for eight weeks significantly improved depression-like behavior by upregulating BDNF levels in patients with depression compared to the placebo group [174].

##### *Lactobacillus* *plantarum*

Many *Lactobacillus plantarum* strains have been reported to improve mental health [175]. Extracellular vesicles derived from *Lactobacillus plantarum* (*L*-EV) have been shown to reverse the reduced BDNF expression and block stress-induced depression-like behaviors. Therefore, *L*-EVs exhibit antidepressant-like behavior in mice with stress-induced depression by altering the expression of neurotrophic factors in the hippocampus [176,177]. Treatment with the probiotic *L. plantarum* IS-10506 upregulated the expression of BDNF, brain neurotrophin, and 5-HT in the brain relative to the control group. Therefore, *L. plantarum* IS-10506 regulates the gut–brain axis, which could potentially promote brain development and function [178].

##### *Lactobacillus rhamnosus* zz-1

*Lactobacillus rhamnosus* zz-1 (*L. rhamnosus*) is one of the most commonly used probiotics and is safe, according to the European Food Safety Authority (EFSA) [179]. *L. rhamnosus* supplementation can effectively treat diarrhea, improve immune function, and regulate the gut microbiota. *L. rhamnosus* has been reported to significantly inhibit hormonal release due to the hyperactivity of the HPA axis, reduce CUMS-induced deficits in monoamine neurotransmitters, and improve BDNF/TrkB levels. These alterations have been partially associated with the regulation of the intestinal microenvironment. *L. rhamnosus* restored intestinal damage and improved intestinal inflammation in depressed mice. In summary, this study established that *L. rhamnosus* was effective in preventing depression due to chronic stress, adding new evidence to support the mental benefits of probiotics [180].

Natural products targeting BDNF in the gut–brain axis as antidepressant therapy are mentioned as below in Table 1.

## 3. Discussion

Depression affects physical health and well-being, which ultimately leads to an unhappy life. Hippocrates rightly said that all diseases begin in the gut; our food should be our medicine, and our medicine should be our food [184]. An integrative and current view of classical brain ailments as whole-body situations, including a chief role for the GIT, leads to strategies that target the gut microbiota to provide safe, new, and effective therapeutic options for neuropsychiatric and neurodegenerative diseases such as depression. This exciting concept related to the gut–brain axis is poised for testing in the coming years. BDNF is a vital and novel component of the emerging field of neurorestoration. BDNF’s effect is related to the alteration of the gut microbial community, and it plays a vital role in neurodevelopment, plasticity, and maturation in adults, as well as in developing brains, to ameliorate depression [185]. In addition, previous evidence suggests that BDNF may be involved in depression, such that the expression of BDNF is decreased in depressed patients. BDNF has also been shown to decrease in response to acute and chronic stress. High levels of glucocorticoid hormones and proinflammatory cytokines are two key players in the responses of several types of stress. They have also been associated with decreased BDNF levels and changes in neurogenesis. Therefore, BDNF protects against stress-induced neural damage, and it affects neurogenesis in hippocampal neurons, which is thought to be involved in the induction of depression [86]. However, the extensive validation of the BDNF mechanisms underlying the connection between neuropsychiatric disorders and the gut microbiota is still lacking. Furthermore, certain antidepressants, especially on first use, can cause lightheadedness, sleepiness, and dizziness. Additionally, individuals using antidepressants for a long time have described side effects such as suicidality, withdrawal symptoms, not feeling like oneself, addiction, and weight gain [186]. Therefore, identifying and exploring antidepressant-rich natural products may facilitate the search for new medicines with fewer side effects. Interestingly, developing such natural remedies for depression may be valuable in developed nations, such as Japan and the Republic of Korea, where the frequency of suicide associated with depression is high [187]. On the basis of evidence collected from the literature, we have provided suggestions for readers and researchers regarding the identification of antidepressant-rich natural products that act via the gut–brain axis and target BDNF [6,188,189]. In addition, we highlighted opportunities for the development of new therapeutic and diagnostic options for augmenting existing interventions.

Despite extensive research, the neurobiology of major depressive disorder and gastroenterology remains poorly understood owing to a lack of biomarkers, relatively low rates of heritability, and the heterogeneity of precipitating factors, including stress [190]. Researchers are exploring BDNF targeting via the gut–brain axis to obtain new insights regarding the pathways underlying depression. The disruptions in the gut–brain axis may affect intestinal motility and secretion and induce cellular alterations of the entero-neuroendocrine systems. Keeping the gut–brain axis healthy should alter the types of gut bacteria. Natural products, including polyphenol-rich foods, may improve gut health, which can benefit the gut–brain axis [86]. Therefore, collecting and reviewing the literature in a single manuscript can facilitate the identification of potential antidepressant agents by researchers and pharmaceutical companies.

Figure 2 presents a summary of the natural products with their constituents and the various channels of BDNF. It compiles the information on BDNF’s function in the gut–brain axis. This demonstrates that BDNF is an initiator that, either directly or indirectly, induces changes in the gut via several signaling pathways. Additionally, it has been shown to contribute to conditions involving abnormal brain functioning, such as depression, neuropsychiatric illnesses, neurodevelopmental disorders, Alzheimer’s disease, Parkinson’s disease, and multiple sclerosis. Therefore, more effective targeted medicines with diversified structures and pharmacological activities can be developed for treating brain-related diseases by investigating the intricate mechanisms of the gut–brain axis.

Several studies included in this review focused on BDNF, targeting the gut–brain signaling pathway. However, the exact mechanism of action of BDNF in the gut remains unknown for most natural products [78,91,94,102,112]. Many animal and clinical studies have been conducted to determine the crucial effects of natural products on depression and neuroprotection; however, the constituents of these products remain mainly unknown [155,159,160,162,163]. Future studies are necessary to identify the constituents of Tualang honey, *Semen Sojae* Praeparatum, *Cocos nucifera* husk fiber, and HJDT. Additionally, they need to focus on improving the analysis of the effective components of CBM588 for the control of mental disorders and confirming the mechanism by which CBM588 affects host depression via the microbiota–intestine–brain axis.

Some plants are edible and are safe to eat. Adding such edible plants as natural antidepressants to our daily diet can augment pharmacological approaches for managing depression. For example, purple cauliflower, a key component of salad plates, is a rich source of anthocyanins and exhibits strong antioxidant activity, relieving depression [181]. Curcumin from *Curcuma longa,* the most common ingredient in Indian cuisine, is considered a nontoxic naturally occurring polyphenol, and it has been proven to be effective in the management of neurological and neurodegenerative diseases [110]. Similarly, nicotinamide riboside from cow milk; *Poria cocos*, an edible mushroom; Tualang honey; orange juice; oleanolic acid from olive oil; limonene from *Citrus sinensis*; and lycopene from watermelon are natural antidepressants from common foods. Certain edible and nonedible plants have been proven to be effective natural antidepressants; however, their active components with antidepressant activities need to be explored, along with their detailed signaling pathways in both animals and clinical trials targeting gut–brain axis pathways. Such active components include cyandin-3-glucoside and cyanidin-3-sophoroside-5-glucoside from purple cauliflower, 3-n-butylphthalide from celery, terpenoids, and polysaccharides from *Poria cocos,* 2,3,5,4′-tetrahydroxystilbene-2-*O*-β-d-glucoside, HJDT, *Semen Sojae* Praeparatum, *Bifidobacterium longum*, *Lactobacillus casei*, and *Cinnamomum zeylanicum.* The dose and duration, along with the safety and toxicology profile, of potential agents of any synthetic, semisynthetic, or herbal formulation play a potential role in the pharmacological effect of the drug [191]. Certain natural products and their metabolites are considered safe for the gut–brain axis; however, their dose and duration need to be evaluated further to obtain optimized pharmacological effects. Although EGCG has a potential antidepressant effect, the dose and duration of EGCG treatment need to be explored to achieve the optimal effect. Interestingly, we did not observe severe toxicity for the above-mentioned potential agents. Our study revealed that some plants and their derivatives target BDNF, other pharmacological targets, or the gut microbiota to relieve depression in different experimental models. However, a detailed study needs to be performed for targeting BDNF via the gut–brain axis. For example, baicalein improves cognitive behavior via the remodeling of the gut microbiota by targeting TMAO [192]. However, the involvement of baicalein targeting BDNF in neuroprotein needs to be evaluated. Luteolin-7-*O*-glucuronide reduces depression via BDNF activation; however, further studies are needed on the gut–brain axis or the microbial community of the GIT [94]. By modifying the expression of NF-kB and BDNF, naringenin alleviates depression and anxiety-like behaviors in mice subjected to repeated hypoxic stress; however, using the gut–brain axis experimental parameters, there is a need to determine the involvement of naringenin in the gut–brain axis pathway to reduce depression [97]. Studies need to evaluate the antidepression potency of hesperidin and the corresponding mechanisms based on the NLRP3 inflammatory signaling pathway and the gut–brain axis [93]. As BDNF plays a major role in protecting the brain from depression, many researchers and scientists have investigated other signaling pathways that can modulate BDNF expression. However, this may open new opportunities for further study on whether certain natural products and their metabolites, such as silibinin from *Silybum marianum* and thymol, are using the gut–brain axis pathway [103,104,127]. In addition, we highlight the necessity of future research on the combination of two or three natural metabolites and their potential pharmacological effects.

The evidence from the above study revealed that almost all studies were performed using in vivo experimental models. After evaluating the potency of natural products and their metabolites, further work can be performed to evaluate their potency in humans. Various antidepressants cause severe side effects in children and teenagers. Studies have shown that 1 out of 50 children exhibits suicidal thoughts and worsening of depression [193]. Our review highlights that most products mentioned above are part of our daily food. The consumption of natural antidepressants can facilitate the management of depression without medicines. This approach can avoid a potential dependency on antidepressant medicine, reduce the economic burden on patients, and improve patience compliance. In particular, the high-flavonoid group might achieve a significant boost in the BDNF level, which corresponds with an improvement in depression. Food rich in antidepressant agents can prevent depression by modulating the gut microbiota. However, the literature lacks structure–activity relationship (SAR) studies related to the gut–brain axis. Therefore, SAR studies for agents targeting the gut–brain axis can be an additional focus of future research. Interestingly, while exploring natural products that act on BDNF directly or indirectly, we did not identify any natural product or natural product derivative that naturally contains BDNF. Therefore, further study should be focused on finding a BDNF substitute. This highlights the importance of our review, as it provides current information on identifying and developing natural products that can boost BDNF levels.

In summary, the information in this article can serve as a valuable resource for the speedy development of antidepressant therapeutics that target the gut–brain axis. Incorporating natural products into our regular diets can aid in the battle against depression, ultimately improving the quality of life of individuals with depression.

## 4. Materials and Methods

Comprehensive data collection was performed until mid-July 2022 using discrete electronic databases, including Scopus, Web of Science, ScienceDirect, Google Scholar, and PubMed, to collect reports published in the previous five years. We explored studies on medicinal plants and their constituents that may target BNDF-associated signaling networks and act as neuroprotective therapies. We screened additional reviews and systematic reviews to identify potentially related citations.

## Figures and Tables

**Figure 1 ijms-23-14968-f001:**
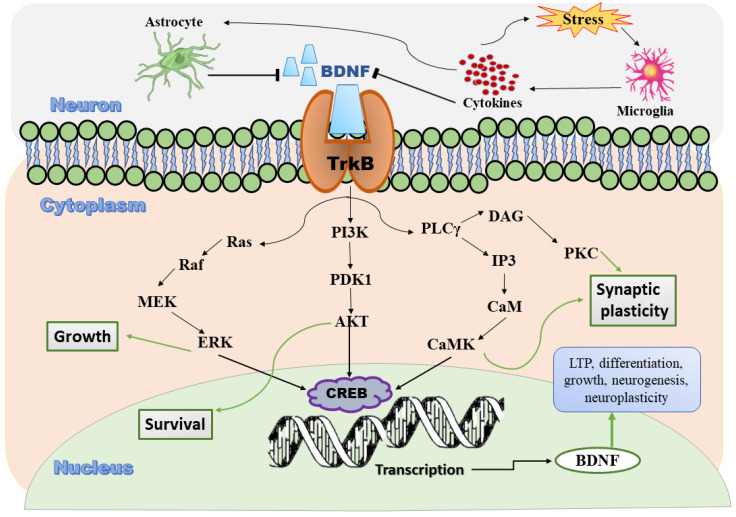
Diagrammatic presentation of signaling pathway of BDNF metabolism. BDNF binds to its high-affinity receptor tyrosine kinase B (TrkB) through the activation of intracellular tyrosine kinase, resulting in the autophosphorylation of TrkB receptors and the sequential activation of three independent cascades. BDNF signaling pathways stimulate one or more transcription factors (cAMP response element binding protein (CREB) and CREB-binding protein) in the nucleus that regulate the expression of genes encoding proteins involved in neural plasticity, differentiation, growth, survival, and stress resistance. The sharp arrow indicates activation, and the T-shaped arrow indicates inhibition. BDNF, brain-derived neurotrophic factor; TrkB, tropomyosin receptor kinase B; PI3K, induction via phosphatidylinositol 3-kinase; PLCγ, phospholipase Cγ; Akt, serine/threonine protein kinase; CaM, calmodulin; CaMK, calcium/calmodulin-dependent protein kinase; CREB, cAMP response element binding protein; DAG, diacylglycerol; ERK, extracellular signal-regulated kinase; IP3, inositol 1,4,5-trisphosphate; MEK, mitogen-activated extracellular signal-regulated kinase; PKC, protein kinase C; PDK1, 3-phosphoinositide-dependent kinase 1.

**Figure 2 ijms-23-14968-f002:**
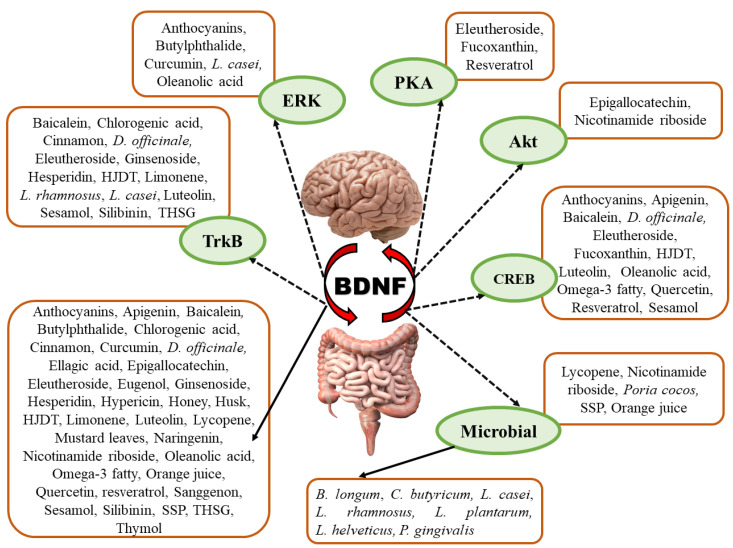
Modulation of BDNF signaling by different natural products. BDNF, brain-derived neurotrophic factor; TrkB, tropomyosin receptor kinase B; Akt, serine/threonine protein kinase; CREB, cAMP response element binding protein; ERK, extracellular signal-regulated kinase; PKA, protein kinase A; HJDT, Huanglian-Jie-Du-Tang; THSG, 2,3,5,4′-tetrahydroxystilbene-2-*O*-β-d-glucoside; SSP, *Semen Sojae* Praeparatum.

**Table 1 ijms-23-14968-t001:** Natural products targeting BDNF in the gut–brain axis as antidepressant therapy.

Natural Products	Sources	Targeted Pathway	Model	Pharmacological Action	References
Cyandin-3-glucoside and cyanidin-3-sophoroside-5-glucoside	Cauliflower (*Brassica oleraceavar.* *Botrytis* L.)	ERK/CREB/ BDNF	Pathogen-free female mice (4 weeks old)	Upregulates BDNF, which also promotes neurogenesis and dendrite development in hippocampus	[181]
Apigenin	Fruits and vegetables	CREB/BDNF	Male Swiss albino mice (12 weeks old)	Prevents cognitive deficit and CREB-BDNF upregulation in the hippocampus	[78]
Baicalein	*Scutellaria* *baicalensis* Georgi	BDNF/TrkB/ CREB	Male C57BL/6J mice (8 weeks old)	Synaptic function protection and neuroprotection against inflammation and depression	[89,90]
3-n-Butylphthalide	Chinese celery	BDNF/ERK/ mTOR and serotonergic	Male Sprague–Dawley (SD; 6 weeks old)	Restores the levels of BDNF, p-ERK, and mTOR	[165]
Chlorogenic acid	Coffee beans	BDNF/TrkB	Male SD rats (6 weeks old)	Induces anxiolytic behavior, modulates autonomic regulation, and activates hippocampal BDNF signaling	[108]
Cinnamon	*Cinnamomum zeylanicum*	BDNF/TrkB	Thirty-two male Wistar rats	Antidepressant	[157]
Curcumin	*Curcuma longa*	BDNF/ERK	Adult male Wistar (6 weeks old)	Antidepressant	[110,111,112]
Alcohol-soluble polysaccharides	*Dendrobium officinale*	BDNF/TrkB/ CREB	Male ICR mice (6 weeks old) and SD rats (6 weeks old)	Antidepressant-like effect by regulating the gut–brain axis.	[159]
Ellagic acid	*Punica granatum*	BDNF and serotoninergic	Male C57BL/6 mice (6 weeks old)	Antidepressant	[114,115]
Epigallocatechin-3-gallate	Green tea (*Camellia sinensis*)	BDNF, Akt, monoaminergic	Female SD rats (6 weeks old), male C57BL/6J mice	Modulates mood, mitigates anxiety, and improves postmenopausal depression	[116,117]
Eleutheroside E	*Acanthopanax senticosus* HARM	PKA/CREB/ BDNF, CREB/BDNF	Male SD rats (6 weeks old)	Antidepressant and regulates autonomic function	[119,120]
Eugenol	*Rhizoma acori* *graminei*	BDNF	Fifty male ddy mice (6 weeks old)	Antidepressant	[182]
Fucoxanthin		PKA/cAMP, CREB	BV-2 cells	Ameliorates neurodegenerative diseases	[149]
Ginsenoside Rg2	*Panax ginseng*	BDNF/TrkB	Adult male C57BL/6J mice (8 weeks old)	Mitigates anxiety/depression and colitis	[129,130]
Hesperidin	Citrus fruits	BDNF/TrkB	ICR male mice (6–8 weeks old), male NMRI mice (10–11 weeks old)	Reduces CUMS and depression caused by mild traumatic brain injury	[91,92,183]
Hypericin	*Hypericum* *triquetrifolium*	BDNF	Adult male Wistar rats	Neuroprotection against depression and inflammation	[144,146]
Limonene	*Citrus* *Sinensis, C.* *Reticulata*	BDNF/TrkB	155 male Kunming mice at 5 weeks	Antidepressant	[131,132]
Luteolin-7-*O*-glucuronide	*Perilla frutescens*	BDNF/TrkB/ CREB	C57BL/6 male mice (7 weeks old)	Improves depression-like and stress-coping behaviors in sleep deprivation	[94]
Lycopene	*Solanum* *lycopersicum*, *Citrullus lanatus*	BDNF, gut microbiota	Male C57BL/6 mice (8 weeks old)	Improves DSS-induced depression and anxiety-like behavior	[152,153]
Mustard leaves	*Brassica juncea*	BNDF and apoptosis	Male C57BL/6 mice (6 weeks old)	Reduces stress and depression	[122]
Naringenin	Grapes (*Vitis* *vinifera*)	BDNF	Thirty-five male Swiss mice	Reduces stress and depression	[96,97]
Nicotinamide riboside	Milk and yeast	gut microbial pathway, BDNF/AKT	Male C57BL/6J mice (7 weeks old)	Alters intestinal microbiota with microglial activation and exhibits antidepressant activity	[143]
Oleanolic acid	*Olea europaea*	ERK1/CREB/ BDNF	Male ICR mice (6 weeks old)	Antidepressant	[135,136]
Omega-3 fatty acids	Fish oil	BDNF/CREB	Male SD rats	Maintains neuronal function and plasticity, improves impaired learning ability after traumatic brain injury, and exhibits antidepressant activity	[134]
Quercetin	Fruits and vegetables	CREB/BDNF	Male SD rats (8 weeks old)	Antidepressant	[98,99]
Resveratrol	*Polygonum* *cuspidatum*	PKA/CREB/ BDNF	Male adult SD	Exhibits anti-IBS-like effects on depression, anxiety, visceral hypersensitivity, and intestinal motility	[124,125]
Sanggenon G	*Morus alba*	BDNF	Male SD rats (8 weeks old)	Antidepressant	[101]
Sesamol	Sesame oil and seeds (*Sesamum indicum* L.)	BDNF/TrkB/ CREB	C57BL/6J male mice (8 weeks old)	Prevents IBD and associated symptoms of anxiety and depression	[142]
Silibinin	*Silybum marianum*	BDNF/TrkB	Male SD rats (6–8 weeks old), male C57BL/6 mice	Attenuated autophagy and neuroprotection against anxiety and depression	[103,104]
Triterpenoids	*Poria cocos*	microbial	Male SD rats	Antidepressant	[138]
2,3,5,4′-Tetrahydroxystilbene-2-*O*-β-d-glucoside	*Polygonum* *multiflorum*	BDNF/TrkB	C57BL/6J mice (6–8 weeks old)	Antidepressant	[128]
Thymol	*T. vulgaris*	BDNF	Adult female Swiss mice (8–10 weeks old)	Restores short-term memory and improves depression	[127]
	Orange juice	Gut microbiota and BDNF	Young adults	Reduces depression through gut microbiota modulation	[107]
Honey	Tualang honey	BDNF	Female SD rats (8 weeks old)	Exhibits anxiolytic and antidepressant effects	[155]
Huanglian-Jie-Du-Tang	*Coptis chinensis Franch*	BDNF/TrkB/ CREB	Male SD rats	Exhibits antidepressant-like effect	[160]
Husk fiber	*Cocos nucifera*	BDNF	Male Swiss mice (8 weeks old)	Neuroprotection against oxidative stress and depression	[162]
*Semen Sojae* Praeparatum	Mixture of mulberry leaves, lentils, black beans fermented with *bacillus* sp. DU-106, *R. Chinensis* 12	microbial and BDNF	Adult male SD rats	Antidepressant and modulates gut microbiota	[163]
Probiotic	*Bifidobacterium longum* NCC3001	BDNF	Adult patients with a diagnosis of IBS with diarrhea and mild to moderate anxiety and/or depression	Antidepressant and adequate relief of IBS symptoms	[170]
Probiotic	*Clostridium* *butyricum*	BDNF and serotonin	Adult (age 6–8 weeks, 18–22 g) male C57BL/6J mice	Regulates gut microbiota and acts as antidepressant	[166,167]
Probiotic	*Lactobacillus casei*	BDNF/ERK, BDNF/TrkB	Pregnant SD female rats	Antidepressant altering gut microbiota composition, brain monoamines, and oxidative stress	[171,172]
Probiotic	*Lactobacillus helveticus* R0052	BDNF	Patients with mild to moderate melancholic depression	Antidepressant	[174]
Probiotic	*Lactobacillus plantarum*	BDNF	Male C57BL/6J mice (7 weeks old), male Wistar rats (8–12 weeks old)	Antidepressant, stimulates the gut–brain axis, and can potentially promote brain development and function.	[176,177,178]
Probiotic	*Lactobacillus rhamnosus* zz-1	BDNF/TrkB	Male C57BL/6 mice (4 weeks old)	Antidepressant via regulating the intestinal microenvironment	[180]
Probiotic	*Porphyromonas gingivalis*	BDNF	Female C57Bl/6J wild-type mice	Antidepressant	[168]

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
