# Peer review of "Antidepressive Effect of Natural Products and Their Derivatives Targeting BDNF-TrkB in Gut–Brain Axis"

_ijms, 2022, doi:10.3390/ijms232314968_

Round 1

Reviewer 1 Report

Liaqat et al., performed a comprehensive revision to offer an overview of our current knowledge in the effect of different natural compounds on the Brain Derived Neurotrofic Factor (BDNF) and their potential implications in the treatment of neuropsychiatric disorders. The topic is interesting and address a major global health problem. Despite that, there are some major concerns that need to be addressed:

1. It would be interesting to indicate the search date of the articles. It would also be necessary to specify the publication date and whether there where a quality assessment for article selection. At least, a global information about the quality of the included studies would be informative.

2. This review includes more than 200 references. Thus there is a huge amount of information and some sections of the review are too extensive. A synthesis effort summarizing and highlighting  the major ideas would provide an easier reading. This would be of special importance for sections 2.1-2.3 and 2.4-2.5.

Author Response

Thank you for all your valuable suggestions and feedback. We have revised the manuscript under consideration of all of the comments from the reviewers and also done some additional changes.All changes are highled with red color in revised mnauscript.

Reviewer 1:

  1. It would be interesting to indicate the search date of the articles. It would also be necessary to specify the publication date and whether there where a quality assessment for article selection. At least, global information about the quality of the included studies would be informative.

Response: Thank you. We have added the searching date (page-26; line: 1009-1011) but the quality assessment did not perform for this review because its an important part of the systematic review. As we already elaborated that we included the articles on medicinal plants and their constituents that may target BNDF-associated signaling networks and act as neuroprotective therapies, especially for depression (page-26; line: 1012-1014).

  1. This review includes more than 200 references. Thus there is a huge amount of information and some sections of the review are too extensive. A synthesis effort summarizing and highlighting  the major ideas would provide an easier reading. This would be of special importance for sections 2.1-2.3 and 2.4-2.5.

Response:

Thank you for your suggestion. We have summarized the introduction part and the above-mentioned points of the manuscript (page-1-8; line: 26-394).

Reviewer 2 Report

The format of the manuscript is unclear. On the one hand, it should be a review and on the other hand, the data were sometimes presented like in a “study”. 

e.g.: in the abstract: 

“This  study  aimed  to  elucidate  the  anti-depressive  effects  of  brain-derived  neurotrophic factor (BDNF)-targeting therapeutic natural products…..” and

“This comprehensive review is an effort to enumerate the effect of numerous natural products…”

The paragraphs of the manuscript should completely be rearranged and redundancies should be avoided.

It is suggested to completely revise the whole manuscript and to shorten the introduction. In the introduction, large passages do not contain references.  

Several sentences should be corrected since they do not make sense in the current version

as e.g. 

“It is  associated with sundry biological functions, including N-methyl-á´…-aspartate (NMDA) activity, synapse stability, and cholinergic, GABAergic, serotonergic, dopaminergic, and synaptogenesis [10].”

or

“Additionally, highlight the positive aspects of this study related to gut-brain axis-related abnormalities for neuroprotection such as depression through BDNF.”

Currently, the structure of the Result section is the following:

2.1. Brain-derived neurotrophic factor and its clinical applications   

2.2. Signaling pathway of BDNF

2.3. BDNF: a novel synaptic repair signaling pathway

2.4. Role of gut microbiota in depression   

2.4.1. Gut microbiota–brain axis in stress, depression, and anxiety

2.4.2. Role of BDNF in IBD-induced depression

2.4.3. Findings related to gut microbiota in patients with depression

2.5. Role of BDNF-mTORC1 signaling pathway in depression

2.6. Effects of antidepressants on BDNF and their related side effects

2.7. Modulation of BDNF signaling by different natural products and their derivatives

The first section of the Results has the heading:

“Brain-derived neurotrophic factor and its clinical applications “However, there is nothing described concerning the “clinical application of BDNF”.

Chapter 2.2 ends with the sentence “BDNF binding to TrkB can regulate at least three intracellular signaling pathways”. However, the next chapter 2.3 only explains a novel synaptic repair signaling pathway. This is somewhat confusing.

It is suggested to continue the review after the introduction 

with a paragraph on neurotrophins and their generation, followed by a description of neurotrophin receptors; the signaling pathway of neurotrophins (or only BDNF), (potential) role of BDNF in major depression; the role of gut microbiota in major depression; the role of the gut-brain axis in major depression; modulation of BDNF signaling by different natural products and their derivatives: possible impact of these products on the gut-brain axis (how these products might enter the CNS? E.g. do these products cross the blood-brain barrier or how they may exert their actions in the CNS); effects of these products on brain BDNF and major depression. The review may end with a conclusion and /or outlook.

Please introduce abbreviations as e.g. DNF; IBD, SPF……

Author Response

Thank you for all your valuable suggestions and feedback. We have revised the manuscript under consideration of all of the comments from the reviewers and also done some additional changes.All changes are highled with red color in revised mnauscript.

Reviewer 2:

  1. The format of the manuscript is unclear. On the one hand, it should be a review and on the other hand, the data were sometimes presented like in a “study”. 

e.g.: in the abstract: “This  study  aimed  to  elucidate  the  anti-depressive  effects  of  brain-derived  neurotrophic factor (BDNF)-targeting therapeutic natural products…..” and

“This comprehensive review is an effort to enumerate the effect of numerous natural products…”

 Response:

Thank you. We have modified these points (page-1; line:12-16; 18-19).

The paragraphs of the manuscript should completely be rearranged and redundancies should be avoided.

Response: Thank you. We have improvised and summarized the whole manuscript.

It is suggested to completely revise the whole manuscript and to shorten the introduction. In the introduction, large passages do not contain references.  

Several sentences should be corrected since they do not make sense in the current version

as e.g. “It is  associated with sundry biological functions, including N-methyl-á´…-aspartate (NMDA) activity, synapse stability, and cholinergic, GABAergic, serotonergic, dopaminergic, and synaptogenesis [10].”

or “Additionally, highlight the positive aspects of this study related to gut-brain axis-related abnormalities for neuroprotection such as depression through BDNF.”

Response: Thank you. We have modified these points (page-3; line: 104-110).

Currently, the structure of the Result section is the following:

2.1. Brain-derived neurotrophic factor and its clinical applications   

2.2. Signaling pathway of BDNF

2.3. BDNF: a novel synaptic repair signaling pathway

2.4. Role of gut microbiota in depression;   

2.4.1. Gut microbiota–brain axis in stress, depression, and anxiety

2.4.2. Role of BDNF in IBD-induced depression

2.4.3. Findings related to gut microbiota in patients with depression

2.5. Role of BDNF-mTORC1 signaling pathway in depression

2.6. Effects of antidepressants on BDNF and their related side effects

2.7. Modulation of BDNF signaling by different natural products and their derivatives

The first section of the Results has the heading:

“Brain-derived neurotrophic factor and its clinical applications “However, there is nothing described concerning the “clinical application of BDNF”.

Response: Thank you. We have modified these points (page-3; line:112).

Chapter 2.2 ends with the sentence “BDNF binding to TrkB can regulate at least three intracellular signaling pathways”. However, the next chapter 2.3 only explains a novel synaptic repair signaling pathway. This is somewhat confusing.

Response: Thank you. We have modified these points (page-4, line: 171-182).

It is suggested to continue the review after the introduction 

with a paragraph on neurotrophins and their generation, followed by a description of neurotrophin receptors; the signaling pathway of neurotrophins (or only BDNF), (potential) role of BDNF in major depression; the role of gut microbiota in major depression; the role of the gut-brain axis in major depression; modulation of BDNF signaling by different natural products and their derivatives: possible impact of these products on the gut-brain axis (how these products might enter the CNS? E.g. do these products cross the blood-brain barrier or how they may exert their actions in the CNS); effects of these products on brain BDNF and major depression. The review may end with a conclusion and /or outlook.

Response:  Thank you for your suggestions. We have modified, merged, and edited the following points while considering your instructions (page-3; line 112, page-4; line: 178, page-6; line: 232, 262, page-7; line: 310).

Please introduce abbreviations as e.g. DNF; IBD, SPF……

Response: Thank you. We have edited the following abbreviations (page-4; line 194, page-6; line 282, page-7; line: 317).

Round 2

Reviewer 1 Report

The authors have properly addressed all my major concerns.

Reviewer 2 Report

The authors have made a fine job of increasing the quality of the manuscript.

The English language and style are improved. Nevertheless, minor spell checking is required.